# ANALYSIS OF GENERALIZATION CAPACITIES OF NEURAL ORDINARY DIFFERENTIAL EQUATIONS AND RESIDUAL NEURAL NETWORKS

## ABSTRACT

Neural ordinary differential equations (neural ODEs) represent a widely-used class of deep learning models characterized by continuous depth. Understanding the generalization error bound is important to evaluate how well a model is expected to perform on new, unseen data. Earlier works in this direction involved considering the linear case on the dynamics function (a function that models the evolution of state variables) of Neural ODE Marion (2024). Other related work is on bound for Neural Controlled ODE Bleistein & Guilloux (2023) that depends on the sampling gap. We consider a class of neural ordinary differential equations (ODEs) with a general nonlinear function for time-dependent and time-independent cases which is Lipschitz with respect to state variables. We observed that the solution of the neural ODEs would be of bound variations if we assume that the dynamics function of Neural ODEs is Lipschitz continuous with respect to the hidden state. We derive a generalization bound for the time-dependent and time-independent Neural ODEs. Using the fact that Neural ODEs are limiting cases of time-dependent Neural ODEs we obtained a bound for the residual neural networks. We showed the effect of overparameterization and domain bound in the generalization error bound. This is the first time, the generalization bound for the Neural ODE with a more general non-linear function has been found.

## 1 INTRODUCTION

Neural Ordinary Differential Equations (Chen et al. (2018)) are a class of deep learning models where the transformation between layers is treated as a continuous process defined by an ordinary differential equation (ODE). This idea generalizes the concept of residual networks (ResNets), where the evolution of the hidden state $z(t)$ over time is modeled by a differential equation

$$\frac{dz(t)}{dt} = f(z(t), t, \theta(t)) \quad \text{with} \quad z(0) = x, \tag{1.1}$$

where $\theta(t)$ represents the parameters of the model.

Unlike discrete representations from the conventional methods, neural ordinary differential equations (Neural ODEs) directly learn continuous latent representations (or latent states) based on a vector field parameterized by a neural network. Kidger et al. (2020) introduced neural controlled differential equations (Neural CDEs), which are continuous-time analogs of ResNets that use controlled paths to represent irregular time series. Neural ODEs are also extended to neural stochastic differential equations (Neural SDEs) with a focus on aspects such as gradient computation, variational inference for latent spaces, and uncertainty quantification. In neural stochastic ODEs (neural SDEs, Oh et al. (2024)), usually a diffusion term is incorporated but a careful design of drift and diffusion term is essential.

With neural ODEs, generally, it is difficult to handle irregular time-series data. Neural controlled differential equations (Kidger et al. (2020)) generalize neural ODEs by incorporating a control mechanism, allowing them to model the evolution of hidden states as controlled differential equations. Studying the statistical properties of neural ODEs is not a trivial task. Since standard measures of statistical complexity in neural networks, such as those discussed by Bartlett et al.

(2019), typically increase with depth, it is unclear why models with effectively infinite depth, like neural ODEs, would demonstrate strong generalization capabilities.

Marion (2024) studied the statistical properties of a class of time-dependent neural ODEs described by the following equation:

$$\frac{d\mathbf{H}_t}{dt} = \mathbf{W}_t \sigma(\mathbf{H}_t),$$

where $\mathbf{W}_t \in \mathbb{R}^{d \times d}$ is a weight matrix that depends on the time index $t$, and $\sigma : \mathbb{R} \to \mathbb{R}$ is an activation function applied component-wise. The model considered by Marion (2024) does not include the case where there are weights inside the non-linearity since they assume the dynamics at time $t$ to be linear with respect to the parameters.

**Contribution**. For a general class of well-posed neural ODEs, where the neural network involves the non-linear weights within the function, there is no result related to the generalization bound. In this work, we consider a Neural ODE model parameterized by $\theta(t)$ of the following form:

$$\frac{dz(t)}{dt} = f(z(t), t, \theta(t)) \quad \text{with} \quad z(0) = x, \tag{1.2}$$

where $f : \mathbb{R}^d \times \mathbb{R}^d \to \mathbb{R}^d$ and $z(t) : [0, L] \to \mathbb{R}^d$. We provide a generalization bound for the large class of parameterized ODEs instead of a linear class, the bound we provided here will hold for a linear class as well and is stricter than the earlier bounds for the linear class of functions. To the best of our knowledge, this is the first available bound for neural ODEs for this class of functions.

**Organization.** Section 1 is devoted to the introduction, and in Section 2, we discuss the realted works. In section 3, we discuss some of the preliminaries and definitions that are crucial for understanding the problem setup. In section 4, we formulated the problem statement and section 5 is devoted to derive results related to generalization bounds. We showed applications to residual neural networks in section 6 and performed numerical experiments in section 7. In the end, some concluding remarks are given in section 8.

## 2 RELATED WORKS

**Hybridizing deep learning and differential equations.** The fusion of deep learning with differential equations has recently garnered renewed interest, although the concept has been explored since the 1990s Rico-Martinez et al. (1992; 1994). A notable advancement was presented by Chen et al. (2018), where they introduced a model that learns a representation $\mathbf{u} \in \mathbb{R}^n$ by setting the initial condition $z(0) = \phi_{\theta(t)}(\mathbf{u})$ for the following ordinary differential equation (ODE):

$$\frac{dz(t)}{dt} = f(z(t), t, \theta(t))$$

where both $f$ and $\phi_{\theta(t)}$ are neural networks. The solution at the final time $t_1$, denoted $\mathbf{z}(t_1)$, is then utilized as input to a conventional machine learning model. This approach seamlessly integrates neural networks and ODEs, offering a robust framework for learning complex dynamical systems. Since then, several works have built on this idea, including theoretical advancements and practical applications as seen in Dupont et al. (2019); Chen et al. (2019; 2020); Finlay et al. (2020). For a more comprehensive overview, readers may refer to the reviews by Massaroli et al. (2020) and Kidger (2022), which delve into the intersection of differential equations and deep learning.

**Generalization of Neural Controlled Differential Equations.** Bleistein & Guilloux (2023) used a Lipschitz-based argument to obtain a sampling-dependant generalization bound for neural controlled differential equations (NCDEs). The NCDE considered was of the following form:

$$dz_t = G_\psi(z_t) d\tilde{x}_t,$$

where $z_t \in \mathbb{R}^p$, and $G_\psi : \mathbb{R}^p \to \mathbb{R}^{p \times d}$ is neural network parametrized by $\psi$, also $\tilde{x}_t \in \mathbb{R}^p$ is continuous path. In this work, it is assumed that $(x_t)$ is Lipschitz which implies that $x = (x_t)_{t \in [0,1]}$ is of bounded variation. They also analyzed how approximation and generalization are affected by irregular sampling.

**Generalization bounds for neural networks.** Bartlett et al. (2017a) derived a margin-based multiclass generalization bound for neural networks that scales with margin-normalized spectral complexity, involving the Lipschitz constant (the product of the spectral norms of the weight matrices) and a correction factor. Long & Sedghi (2019) established generalization error bounds for convolutional networks based on training loss, parameter count, the Lipschitz constant of the loss, and the distance between current and initial weights, independent of input size and hidden layer dimensions. Experiments on CIFAR-10 show these bounds align with observed generalization gaps under varying hyperparameters in deep convolutional networks. Wang & Ma (2022) derive generalization error bounds for deep neural networks trained via SGD by combining control of parameter norms with Rademacher complexity estimates. These bounds, which apply to various architectures like MLPs and CNNs, depend on the training loss and do not require L-smoothness, making them more broadly applicable than stability-based bounds.

## 3 PRELIMINARIES

**Definition 3.1** (Bartlett et al. (2017b)). *Let $(M, \rho)$ be a metric space. A subset $\hat{T} \subseteq M$ is called an $\tau$-cover of $T \subseteq M$ if for every $m \in T$, there exists an $m' \in \hat{T}$ such that $\rho(m, m') \leq \tau$. $\hat{T}$ is called a proper cover if $\hat{T} \subset T$. The $\tau$ covering number of $T$ is the cardinality of the smallest $\tau$-cover of $T$, that is*

$$N(\tau, T, \rho) = \min\{|\hat{T}| : \hat{T} \text{ is an } \tau \text{ cover of } T\}.$$

**Definition 3.2** (Dutta & Nguyen (2018)). *The function $u \in L^1(\Omega, \mathbb{R})$ is a function of bounded variation on $\Omega$ (denoted by $BV(\Omega, \mathbb{R})$) if the distributional derivative of $u$ is representable by a finite Radon measure in $\Omega$, i.e., if*

$$\int_\Omega u \cdot \frac{\partial \varphi}{\partial x_i} \, dx = -\int_\Omega \varphi \, dD_i u \quad \text{for all } \varphi \in C_c^1(\Omega, \mathbb{R}), \ i \in \{1, 2, \ldots, n\},$$

*for some Radon measure $Du = (D_1 u, D_2 u, \ldots, D_n u)$. We denote by $|Du|$ the total variation of the vector measure $Du$, i.e.,*

$$|Du|(\Omega) \quad = \sup \left\{ \int_\Omega u(x) \, div(\phi) \, dx \, \bigg| \, \phi \in C_c^1(\Omega, \mathbb{R}^n), \ \|\phi\|_{L^\infty(\Omega)} \leq 1 \right\}.$$

**Lemma 3.3** (Gautschi (1959)). *For $x > 0$ and $0 < \lambda < 1$, the inequality holds*

$$x^{1-\lambda} \leq \frac{\Gamma(x+1)}{\Gamma(x+\lambda)} \leq (x+1)^{1-\lambda}.$$

**Rademacher Complexity :** Rademacher complexity is a concept from statistical learning theory that measures the richness of a class of functions in terms of how well they can fit random noise. It is commonly used to derive bounds on the generalization error of learning algorithms.

**Definition 3.4** (Mohri (2018)). *Given a class of functions $\mathcal{H}$ mapping from an input space $\mathcal{X}$ to $\mathbb{R}$ and a sample $S = \{x_1, x_2, \ldots, x_n\}$ drawn from a distribution $\mathcal{D}$, the empirical Rademacher complexity of $\mathcal{H}$ with respect to the sample $S$ is defined as:*

$$\hat{\mathcal{R}}_S(\mathcal{H}) = \mathbb{E}_\sigma \left[ \sup_{h \in \mathcal{H}} \frac{1}{n} \sum_{i=1}^n \sigma_i h(x_i) \right],$$

*where $\sigma_i$ are independent Rademacher variables, which take values $+1$ or $-1$ with equal probability. and the expectation $\mathbb{E}_\sigma$ is taken over the distribution of the Rademacher variables.*

**Lemma 3.5.** *(Particular case of Gronwall's Inequality) Let $I$ denote an interval of the real line of the form $[a, \infty)$ or $[a, b]$ or $[a, b)$ with $a < b$. Let $\alpha$, $\beta$, and $\mathfrak{u}$ be real-valued functions defined on $I$. Assume that $\beta$ and $\mathfrak{u}$ are continuous and that the negative part of $\alpha$ is integrable on every closed and bounded subinterval of $I$.*

*If $\beta$ is non-negative and if $\mathfrak{u}$ satisfies the integral inequality and if the function $\alpha$ is non-decreasing, then*

$$u(t) \leq \alpha(t) + \int_a^t \beta(s)u(s)\,ds, \qquad \forall t \in I,$$

*then*

$$u(t) \leq \alpha(t) \exp\left(\int_a^t \beta(s)\,ds\right), \qquad t \in I.$$

**Lemma 3.6.** *(Gronwall's Lemma for sequences). Let $(y_k)_{k \geq 0}$, $(b_k)_{k \geq 0}$, and $(f_k)_{k \geq 0}$ be positive sequences of real numbers such that*

$$y_n \leq f_n + \sum_{l=0}^{n-1} b_l y_l$$

*for all $n \geq 0$. Then*

$$y_n \leq f_n + \sum_{l=0}^{n-1} f_l b_l \prod_{j=l+1}^{n-1} (1 + b_j)$$

*for all $n \geq 0$.*

Proof can be found in Holte (2009) and Clark (1987). We need a variant of Gronwall's Lemma for sequences.

**Lemma 3.7.** *Let $(u_k)_{k \geq 0}$ be a sequence such that for all $k \geq 1$,*

$$u_k \leq a_k u_{k-1} + b_k$$

*for $(a_k)_{k \geq 1}$ and $(b_k)_{k \geq 1}$ two positive sequences. Then for all $k \geq 1$,*

$$u_k \leq \left(\prod_{j=1}^k a_j\right) u_0 + \sum_{j=1}^k b_j \left(\prod_{i=j+1}^k a_i\right).$$

**Lemma 3.8** (Bartlett et al. (2017b)). *For any function class $\mathcal{F}$ containing functions $f : \mathcal{X} \to \mathbb{R}$, we have that*

$$\hat{R}_n(\mathcal{F}) \;\; \leq \;\; \inf_{\epsilon \geq 0} \left\{ 4\epsilon + 12 \int_\epsilon^{\sup_{f \in \mathcal{F}} \sqrt{\mathbb{E}[\hat{f}^2]}} \sqrt{\frac{\log N(\tau, \mathcal{F}, L_2(P_n))}{n}}\,d\tau \right\}$$

*where $N(\tau, \mathcal{F}, L_2(P_n))$ denotes the covering number of $\mathcal{F}$.*

**Definition 3.9.** *Let $z(t)$ be the solution of the neural ODE to 1.1 with $x$ as the initial solution. The empirical risk over the training data is :*

$$\hat{R}(z(t)) = \frac{1}{n} \sum_{i=1}^n \ell(y_i, z(t)).$$

*The expected risk or generalization error over the data distribution is :*

$$R(z(t)) = \mathbb{E}_{(x,y) \sim \mathcal{P}}[\ell(y, z(t))].$$

$$R(z(t)) = \mathbb{E}[\ell(y, z(t))].$$

**Lemma 3.10** (Mohri (2018)). *Rademacher complexity regression bounds : Let $L : \mathcal{Y} \times \mathcal{Y} \to \mathbb{R}$ be a non-negative loss function, upper bounded by $M > 0$ ($\ell(y, y') \leq M$ for all $y, y' \in \mathcal{Y}$), and such that for any fixed $y' \in \mathcal{Y}$, the function $y \mapsto \ell(y, y')$ is $\mu$-Lipschitz for some $\mu > 0$.*

$$\mathbb{E}_{(x,y) \sim \mathcal{D}}[\ell(h(x), y)] \;\; \leq \;\; \frac{1}{n} \sum_{i=1}^n \ell(h(x_i), y_i) + 2\mu \hat{\mathcal{R}}_S(\mathcal{H}) + 3M \sqrt{\frac{\log \frac{2}{\delta}}{2n}}.$$

## 4 THE LEARNING PROBLEM

The evolution of the hidden state $z(t)$ over time is modeled by a differential equation

$$\frac{dz(t)}{dt} = f(z(t), t, \theta(t)) \quad \text{with} \quad z(0) = z_0, \tag{4.3}$$

We now detail our learning setup. Let $z(t)$ be the solution of Neural ODE and let $y_i$ be the true label of the differential equation at $i^{th}$ time step to be learned by Neural ODE.

We consider an i.i.d. sample $\{(y_i, t_i)\}_{i=1}^n \sim y, t$. For a given predictor $z(t) \in \mathcal{F}$, define

$$R^n(z(t)) = \frac{1}{n} \sum_{i=1}^n \ell(y_i, z(t_i)) \quad \text{and} \quad R(z(t)) = \mathbb{E}_{t,y}\left[\ell(y, z(t))\right]$$

as the empirical risk and expected risk on the continuous data. $R^n(z(t))$ cannot be optimized, since we do not have access to the continuous data. Let $\hat{\theta}(t) \in \arg\min_{\theta(t)} \in \Theta(t) R^n(z(t))$ be an optimal parameter and $\hat{z}(t)$ be the optimal predictor obtained by empirical risk minimization. In order to obtain generalization bounds, the following assumptions on the loss and the outcome are necessary Mohri (2018).

**Assumption 1.** $f(z(t), t, \theta(t))$ is assumed to be Lipschitz continuous with respect with $z(t)$.

**Assumption 2.** Weights $A_{i(t)}$ and biases $b_{i(t)}$ are Lipschitz continuous.

**Assumption 3.** The outcome $y \in \mathbb{R}^d$ is bounded almost surely.

**Assumption 4.** The loss $\ell : \mathbb{R}^d \times \mathbb{R}^d \to \mathbb{R}_+$ is Lipschitz continuous with respect to its second variable, that is, there exists $L_\ell$ such that for all $u, u' \in Y$ and $y \in Y$,

$$|\ell(y, u) - \ell(y, u')| \le L_\ell |u - u'|.$$

This hypothesis is satisfied for most of the classical loss functions, such as the mean squared error, as long as the outcome and the predictions are bounded. This is true by Assumption 3 and Lemma 5.1. The loss function is thus bounded since it is continuous on a compact set, and we let $M_\ell$ be a bound on the loss function.

## 5 MAIN RESULTS

We state and prove important lemmas before proceeding to the proof of the main theorem 5.9. We assume that $f(z(t), t, \theta(t))$ is Lipschitz continuous with respect to $z$. So, by the mean value theorem, the solution to equation 1.1 will be of bounded variation.

**Lemma 5.1.** *For* $z(t) \in \mathbb{R}^d, A_i(t) \in \mathbb{R}^{m \times d}$ *and* $b_i(t) \in \mathbb{R}^d$ *for* $i = 1, 2 \ldots N$

$$f_N(z(t)) \quad := \quad \sigma\left(A_N(t)\sigma\left(A_{N-1}(t)\sigma\left(\ldots \sigma\left(A_1(t)z + b_1\right)\right) + b_{N-1}\right) + b_N\right).$$

*Assume that $\sigma$ is $L_\sigma$ Lipschitz, and $A_i$'s are bounded by $\mathcal{A}$ and biased terms are bounded by $\mathbf{B}$. Let $\|A_i(0)\| \le B_{A_0}$, $\|b_i(0)\| \le B_{b_0}$, $t \in [0, L]$ and $L_A$ and $L_b$ are Lipschitz constant for weights and biases respectively.*

*Using equation A.5 we have, $\|A_i(t)\| \le \|A_i(0)\| + L_A L \le B_{A_0} + L_A L = \mathcal{A}$ and $\|b_i(t)\| \le \|b_i(0)\| + L_b L \le B_{b_0} + L_b L = \mathbf{B}$. Then,*

$$\|z(t)\| \le \left(\|z(0)\| + tL_\sigma \mathbf{B}\frac{(L_\sigma \mathcal{A}^N - 1)}{L_\sigma \mathcal{A}}\right) \exp\left(tL_{f_{\theta(t)}}\right).$$

**Corollary 5.2.** *In the case of time-independent Neural ODE Lipschitz constants for weights and biases will be 0. hence $B_{A_0} = \mathcal{A}$ and $B_{b_0} = \mathbf{B}$ and*

$$\|z(t)\| \le \left(\|z(0)\| + LL_\sigma B_{b_0}\frac{(L_\sigma B_{A_0}^N - 1)}{L_\sigma B_{A_0}}\right) \exp\left(LL_{f_{\theta(t)}}\right). \tag{5.4}$$

*This bound on the solution will be useful to obtain the explicit form for covering number bound. This bound involves the Lipschitz constant, bound on biased terms and weights.*

**Lemma 5.3.** *Let*

$$V = \left( \|z(0)\| + LL_\sigma \mathbf{B} \frac{(L_\sigma \mathcal{A}^N - 1)}{L_\sigma \mathcal{A}} \right) \exp \left( LL_{f_{\theta(t)}} \right)$$

*and* $0 < \tau \leq \frac{LV}{\tau}$, *then*

$$N(\tau, \mathcal{I}, L_2(P_n)) \leq \frac{2^{4\frac{LV}{\tau}}}{18}.$$

**Remark 5.4.** *Observe that the covering number bound increases exponentially with domain size and the bound of solution. We obtain a strict bound on covering number for the class of $L^1$ functions.*

**Corollary 5.5.** *Let* $0 < \tau \leq \frac{LV}{\tau}$, *then*

$$N(\tau, \mathcal{B}, L_2(P_n)) \leq \frac{2^{16\frac{LV}{\tau}}}{324}.$$

*Proof.* From Dutta & Nguyen (2018), we know that

$$N(\tau, \mathcal{B}, L_2(P_n)) \leq N^2 \left( \frac{\tau}{2}, \mathcal{I}, L_2(P_n) \right),$$

which proves the required result. $\qquad\square$

**Remark 5.6.** *In the above lemma, the bound is dependent on covering number of non-decreasing functions with the radius of balls getting half. But for the class of bounded variation functions, we do not assume that the functions are non-decreasing.*

**Lemma 5.7.** *Let $\mathcal{B}'$ be the class of $\mathbb{R}^d$ valued functions with domain $[0, L]$ that are of bounded variation, then*

$$\hat{\mathcal{R}}_n(\mathcal{B}') \leq 96 \frac{\sqrt{bLVd^{\frac{3}{2}} \log 2}}{\sqrt{n}} - 576 \frac{LVd^{\frac{3}{2}} \log 2}{n}.$$

**Remark 5.8.** *Lemma 5.7 ensures that the bound on Rademacher complexity increases with the dimension of range space for bounded variation functions. Also, due to the constant $V$, we also get the dependence on weight parameters and Lipschitz constant of activation functions.*

**Theorem 5.9.** *(Generalization bound for Neural ODEs) Let $V$ be the upper bound of the solution of neural ODE, $d$ be the dimension of the solution, $\hat{z}(t)$ be the optimal predictor and $z^*(t)$ be the true solution and $L$ be the upper bound for time and $M > 0$ be an upper bound of non-negative loss function $l : [0, V] \times [0, V] \to \mathbb{R}$, i.e., $l(\hat{z}(t), z^*(t)) \leq M$ for all $\hat{z}(t), z^*(t) \in [0, V]$. Also, assume that for any fixed $\hat{z}(t) \in [0, V]$, the mapping $y \mapsto l(\hat{z}(t), z^*(t))$ is $\mu$-Lipschitz for some $\mu > 0$. Then generalization error is bounded with probability at least $1 - \delta$ by:*

$$R(\hat{z}(t)) \quad \leq \quad R^n(\hat{z}(t)) + 2\mu \left( 96 \frac{\sqrt{bLVd^{\frac{3}{2}} \log 2}}{\sqrt{n}} - 576 \frac{LVd^{\frac{3}{2}} \log 2}{n} \right) + 3M \sqrt{\frac{\log \frac{2}{\delta}}{2n}}.$$

**Outline of the proof.** We observed that the solution of the Neural ODEs described by equation 1.1 will be of bounded variations. We found stricter bound for covering number of this class of functions. We observed that the covering number is related to the number of positive integer solutions of an equation which is equal to central binomial coefficients. The central binomial coefficient obeys a recurrence relation which has a closed form solution. We then used inequality which the ratio of gamma functions satisfies. In this way, we obtained a stricter bound for the covering number of bounded variation functions. We assumed the parameters to be Lipschitz continuous and obtained a bound on Weights and biases. We then found Rademacher complexity bound using Dudley's entropy integral stated in Lemma 3.8. Finally, we used the result for the Rademacher complexity in Lemma 5.7 to regression bound stated in Lemma 3.10. Since Rademacher complexity is non negative, $b \geq \frac{36LV \log 2}{n}$.

**Comparison:** The bound given in our work is stricter in terms of $n$. Bound given in Marion (2024) does not depend on depth but has worse dependence on width, our bound depends on depth but

does not depend on width. In the bound given in Bleistein & Guilloux (2023), if we take the case $x(t) = t$, in which case it is neural ODE, the bound is the same in terms of $n$ but the bound depends on the discretization of time, here it is independent of that and also the bound is simpler in this case as it contains less number of parameters. More details can be found in appendix A.4.

## 6 APPLICATION TO RESIDUAL NEURAL NETWORKS

Residual Neural Networks (ResNets) operate on discrete sequences of inputs and produce hidden states at each time step. The hidden state $h_t$ at time step $t$ is updated using the previous hidden state $h_{t-1}$ and the current input $x_t$. The update rule for a basic ResNet can be written as:

$$h_t = h_{t-1} + \Delta t \cdot f(h_{t-1}, \theta_t)$$

where $h_t$ is the hidden state at time step $t$, $f(h_{t-1}, \theta_t)$ is the update function (a nonlinear transformation parameterized by weights $\theta_t$), and $\Delta t$ is the time step size (assumed to be 1). This discrete update rule resembles the Euler's method for solving ordinary differential equations (ODEs). If we interpret $h_t$ as a function of continuous time $t$, we can write:

$$\frac{dh(t)}{dt} = f(h(t), t, \theta(t)).$$

This equation describes how the hidden state $h(t)$ evolves continuously over time. The right-hand side has the same update function as in the discrete ResNet, but in this case, it governs continuous evolution.

To connect this with Neural ODEs, we take the continuous-time limit of the discrete ResNet. As $\Delta t \to 0$, the discrete ResNet update equation becomes a continuous-time differential equation:

$$\lim_{\Delta t \to 0} \frac{h_{t+\Delta t} - h_t}{\Delta t} = f(h(t), t, \theta(t)).$$

This is exactly the form of a Neural ODE:

$$\frac{dh(t)}{dt} = f(h(t), t, \theta(t)).$$

In this case, $h(t)$ is the continuously evolving hidden state, and $f(h(t), t, \theta(t))$ is the neural network defining the dynamics. The solution to this differential equation gives the evolution of the hidden state over continuous time, instead of discrete steps as in ResNets. Hence, Neural ODEs can be seen as the limiting case of ResNets when the time step between updates goes to zero, allowing the hidden state to evolve continuously.

**Theorem 6.1.** *(Generalization bound for Residual Neural Networks) Let $V$ be the upper bound of the solution of ResNet, $d$ be the dimension of the solution, $\hat{z}(t)$ be the optimal predictor and $z^*(t)$ be the true solution and $L$ be the upper bound for time and $M > 0$ be an upper bound of non-negative loss function $l : [0, V] \times [0, V] \to \mathbb{R}$, i.e., $l(\hat{z}(t), z^*(t)) \leq M$ for all $\hat{z}(t), z^*(t) \in [0, V]$, Also, assume that for any fixed $\hat{z}(t) \in [0, V]$, the mapping $y \mapsto l(\hat{z}(t), z^*(t))$ is $\mu$-Lipschitz for some $\mu > 0$. Then generalization error is bounded with probability at least $1 - \delta$ by:*

$$R(\hat{z}(t)) \quad \leq \quad R^n(\hat{z}(t)) + 2\mu \left( 96 \frac{\sqrt{bLVd^{\frac{3}{2}} \log 2}}{\sqrt{n}} - 576 \frac{LVd^{\frac{3}{2}} \log 2}{n} \right) + 3M \sqrt{\frac{\log \frac{2}{\delta}}{2n}}.$$

**Outline of the proof.**

If $L$(bound on time domain) is discrete then the final time $L$ of neural ODE will be same as Number of layers of Resnet, $V$ will hold for any time, it will be same for final time $t$ for Neural ODE and Resnet, $L$ can be real number and since the number of layers is discrete we take $N = \lfloor L \rfloor$. Since $f$ is same for both, $L_f$ will also be same hence $V$ will be same. $N$ will always be less than or equal to $L$, hence covering number for set of solutions by Resnet will be less than or equal to covering number for set of solutions by Neural ODE. If we use Dudley's entropy integral then Rademacher complexity bound will be the same because covering the number of set of hidden states of ResNet will be less than or equal to covering number of set of solutions of Neural ODEs hence if we substitute this

Rademacher complexity bound to the regression bound by Mohri (2018) then we get same bound.

**Comparison:** The bound obtained by us depends on depth only, but the bound given in Marion (2024) depends on width and depth. More details can be found in appendix A.5.

# 7   NUMERICAL ILLUSTRATIONS

The experiment shown by figure 1 investigates the impact of varying the number of hidden units in a Neural ODE model on the generalization error. Neural ODEs are continuous-depth neural networks that model the dynamics of a system using ordinary differential equations. In this setup, we train a Neural ODE with a hidden layer whose dimension is altered across different experiments. We utilize a simple two-dimensional input and a synthetic dataset where the target is generated by applying a sine function to the sum of the input features. The network's task is to predict the scalar output corresponding to this target. By adjusting the number of hidden units in the ODE block, we analyze how the model's capacity affects its generalization ability, defined as the difference in error on unseen test data after training. Overall as the number of hidden units increases the generalization error increases which validates the theorem 5.9, because as the number of hidden units increases the norm bound increases.

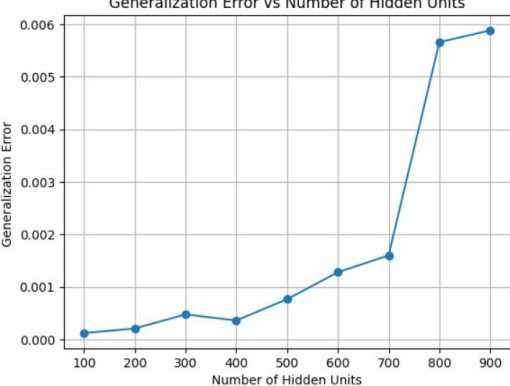

Figure 1: Generalization Error vs Number of Hidden Units in Neural ODE.

In the experiment illustrated by figure 2, we utilized a deeper Neural ODE model to assess the effect of different regularization parameters on the generalization gap. The synthetic data simulates complex real-life phenomena, such as a particle trajectory in a potential field. Regularization was applied to the model's loss function to enforce model stability, and the regularization term $V$ was computed as a function of various properties of the model, such as the spectral norms of weight matrices. For each trial, we recorded the generalization gap as the difference between training and test losses. The results were visualized using a box plot, which shows the distribution of generalization gaps for each regularization parameter, providing insights into the bound which is directly proportional to $V$. As we increase the value of the regularization parameter the mean generalization gap decreases which indicates that the bound is directly proportional to $V$. The constant $V$ is dependent on the bound of weights and bias terms which changes for each training.

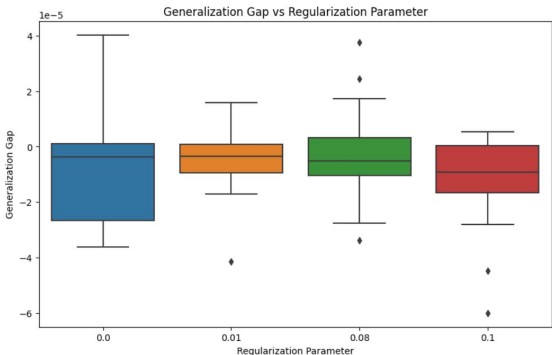

Figure 2: Plot of generalization gap against regularization parameter for time-independent Neural ODE. $V$ which is the bound of the solution is added as a penalty term to the loss function. For each value of the regularization parameter Neural ODE, 20 trials were done. For each trial, it was trained for 25 epochs.

The experiment shown by figure 3 is to investigate how the generalization gap is related to the Lipschitz constant of weights $\sup_{0 \le k \le L-1} \|W_{k+1} - W_k\|$. The Neural ODE is defined with time-varying weights, where the forward pass involves applying a sinusoidal time dependency to the weights of the hidden layer. The model computes the Lipschitz constant by calculating the largest singular value of the weight matrices, which serves as a measure of how sensitive the model is to input changes. Lipschitz constant of weights is added as a penalty term in the loss function with different regularization parameters ($\lambda$ )values. The results are summarized in a box plot, showing the generalization gap versus $\lambda$, to visualize the impact of varying the penalization factor on the model's generalization performance. As we increase the value of the penalization factor the average generalization gap decreases which indicates models with less Lipschitz constant of weights have less generalization gap.

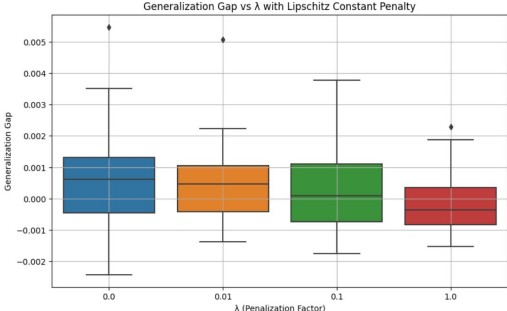

Figure 3: Plot of generalization gap against regularization parameter for time dependent Neural ODE. Lipschitz constant for weights which is the bound of solution is added as a penalty term to the loss function. Four different $\lambda$ values (0, 0.01, 0.1, and 1) are tested over 20 trials. For each trial, the generalization gap is calculated as the difference between the validation loss and training loss after training for 50 epochs.

## 8   CONCLUSION

We obtain the first generalization bounds for time-independent and time-dependent neural ODEs. We proved the generalization bound for the time-dependent neural ODEs of the form $dH_t = f(H_t, \theta(t))dt$. Using this result we also obtain the generalization bound for the time-independent neural ODEs. By extending the reasoning for the time-dependent neural ODEs to the discrete case we obtain generalization bound for residual neural networks. We also showed how it depends on the width and depth of the network by considering the term which depends on the depth and width of the network as it contains the norm which depends on the width of the network. We also showed

how the generalization gap is dependent on the Lipschitz constant of weights for the case of time-dependent Neural ODEs. Since stochastic Neural ODEs have been found to deep limits of a large class of residual neural networks, it will be interesting to extend our result to the more involved case of Neural SDEs.

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

## APPENDIX

**Organization of the Appendix:** Section A in appendix provides the proofs for three lemmas which we used to prove the main theorem. We also provide comparison results in section A. We proved lemma 5.1, 5.3, and 5.7 in this section. Section B is devoted to details related to numerical experiments.

## A   PROOFS

### A.1   PROOF OF LEMMA 5.1

*Proof.* Let $A(t)$ be a time-dependent matrix. We assume that $A(t)$ is *Lipschitz continuous*, meaning there exists a constant $L_A$ such that for all $t_1, t_2 \in [t_0, t_f]$:

$$\|A(t_1) - A(t_2)\| \leq L_A |t_1 - t_2|$$

where $L_A$ is the *Lipschitz constant* and $\| \cdot \|$ is a suitable matrix norm (e.g., Frobenius norm or operator norm).

To express $A(t)$ as a function of its initial value $A(t_0)$, we use the integral representation:

$$A(t) = A(t_0) + \int_{t_0}^t \frac{dA(s)}{ds}\, ds$$

where $\frac{dA(s)}{ds}$ is the time derivative of $A(s)$, and the integral captures the accumulation of changes over time.

Using the assumption that $A(t)$ is Lipschitz continuous, the time derivative $\frac{dA(s)}{ds}$ is bounded by the Lipschitz constant $L_A$. Therefore, for $s \in [t_0, t_f]$, we have:

$$\left\| \frac{dA(s)}{ds} \right\| \leq L_A$$

Substituting this bound into the integral representation of $A(t)$:

$$\|A(t) - A(t_0)\| \leq \int_{t_0}^t \left\| \frac{dA(s)}{ds} \right\| ds \leq \int_{t_0}^t L_A\, ds$$

This simplifies to:

$$\|A(t) - A(t_0)\| \leq L_A |t - t_0|$$

Thus, we have the bound:

$$\|A(t)\| \leq \|A(t_0)\| + L_A |t - t_0|$$

Finally, to remove the time dependency, we maximize the bound over the interval $[t_0, t_f]$:

$$\|A(t)\| \leq \|A(t_0)\| + L_A(t_f - t_0)$$

Thus, the matrix $A(t)$ is uniformly bounded by a time-independent constant $M_A$:

$$\|A(t)\| \leq M_A = \|A(t_0)\| + L_A(t_f - t_0)$$

Since $t \in [0, L]$,

$$M_A = \|A(0)\| + L_A L \tag{A.5}$$

Let

$$f_N(z(t))$$
$$:= \quad \sigma \left( A_N(t) \sigma \left( A_{N-1}(t) \sigma \left( \ldots \sigma \left( A_1(t) z + b_1(t) \right) \right) \right. \right.$$
$$+ b_{N-1}(t) \right) + b_N(t) \big) .$$

where $z \in \mathbb{R}^d$.

Let us first consider the case when d=1.

Let $Dz$ be the distributional derivative of solution function $z$ and

$$\mathcal{I} = \{z \in L^1([0, L]) \mid z \text{ is non decreasing}\}$$

$$\mathcal{B} = \{z \in L^1([0, L]) \mid |Dz|((0, L)) \leq M\}.$$

We know that finding solution to neural ODE (5.1) is equivalent to finding solution to the integral equation

$$z(t) = z(0) + \int_0^t f(z(t), t, \theta(t)\, dt. \tag{A.6}$$

Taking norms, this yields:

$$\|z(t)\| \leq \|z(0)\| + \int_0^t \|f(z(t), t, \theta(t))\| \, dt. \tag{A.7}$$

Notice that since we assumed $f$ is Lipschitz with respect to $z$, we have that for all $z \in \mathbb{R}^d$:

$$\|f(z(t), t, \theta(t))\| \leq \|f(z(t), t, \theta(t)) - f(0, t, \theta(t))\| + \|f(0, t, \theta(t))\| \tag{A.8}$$
$$\leq \|f(z(t), t, \theta(t)) - f(0, t, \theta(t))\| + \|f(0, t, \theta(t))\| \tag{A.9}$$
$$\leq L_f \|z(t)\| + \|f(0, t, \theta(t))\| \tag{A.10}$$

where the last inequality follows from the fact that $f$ is Lipschitz. It follows that:

$$\|z(t)\| \leq \|z(0)\| + \int_0^t \left( L_f \|z\| + \|f(0, t, \theta(t))\| \right) dt. \tag{A.11}$$

Using the fact that $\int_0^t dt = t$, one gets:

$$\|z(t)\| \leq \|z(0)\| + t \|f(0, t, \theta(t))\| + L_f \int_0^t \|z(t)\| \, dt \tag{A.12}$$

Applying Gronwall's inequality stated in Lemma 3.5 yields,

$$\|z(t)\| \leq \left( \|z(0)\| + t \|f(0, t, \theta(t))\| \right) \exp\left( t L_f \right). \tag{A.13}$$

Let $\|A_i(0)\| \leq B_{A_0}$ and $\|b_i(0)\| \leq B_{b_0}$

Then using equation A.5 we get, $\|A_i(t)\| \leq \|A_i(0)\| + L_A L \leq B_{A_0} + L_A L = \mathcal{A}$ and $\|b_i(t)\| \leq \|b_i(0)\| + L_b L \leq B_{b_0} + L_b L = \mathbf{B}$

Since

$$\|f_N(0)\| = \|f_N(0) - \sigma(0)\| \le L_\sigma \|A_N(t)f_{N-1}(0)\| + L_\sigma \mathbf{B}, \tag{A.14}$$

$$\le L_\sigma \mathcal{A} \|f_{N-1}(0)\| + L_\sigma \mathbf{B}. \tag{A.15}$$

Using lemma 3.7,

$$\|f_N(0)\| \le (L_\sigma \mathcal{A})^{N-1} \|\sigma(b_1)\| + L_\sigma \mathbf{B} \sum_{j=0}^{N-2} (L_\sigma \mathcal{A})^j, \tag{A.16}$$

$$\le L_\sigma \mathbf{B} \sum_{j=0}^{N-1} (L_\sigma \mathcal{A})^j, \tag{A.17}$$

$$= L_\sigma \mathbf{B} \frac{(L_\sigma \mathcal{A})^N - 1}{L_\sigma \mathcal{A} - 1}. \tag{A.18}$$

This implies

$$\|z(t)\| \le \left( \|z(0)\| + t L_\sigma \mathbf{B} \frac{(L_\sigma \mathcal{A}^N - 1)}{L_\sigma \mathcal{A}} \right) \exp(t L_f). \tag{A.19}$$

$\square$

## A.2 PROOF OF LEMMA 5.3

*Proof.* For a fixed positive integer $n$, let us set the discretization size as $\Delta x = \frac{L}{n}$, $\Delta y = \frac{V}{n}$. To each $z \in \mathcal{I}$, we associate the pair of functions $(\psi^+[z], \psi^-[z])$ defined by

$$\psi^{\overline{+}}[z] = \sum_{k=0}^{N-1} \psi_k^{\overline{+}} \cdot \mathbf{I}[k \cdot \Delta x, (k+1) \cdot \Delta x], \tag{A.20}$$

where

$$\psi_k^- = \left[ \frac{z(k \cdot \Delta x + 0)}{\Delta y} \right],$$

$$\psi_k^+ = \left[ \frac{z((k+1) \cdot \Delta x - 0)}{\Delta y} \right] + 1.$$

For $\mathcal{X}^{\overline{+}} \in \mathcal{I}$, define

$$U(\mathcal{X}^-, \mathcal{X}^+) = \{z \in \mathcal{I} \mid \mathcal{X}^- \le z \le \mathcal{X}^+\}.$$

Since $z \in U(\mathcal{X}^-[z], \mathcal{X}^+[z])$, the set

$$\mathcal{U} = \{U(\mathcal{X}^-[z], \mathcal{X}^+[z]) \mid f \in \mathcal{I}\}$$

is a covering of $\mathcal{I}$.

Since

$$\#\mathcal{U} \le \{0 \le a_0 \le a_1 \le \cdots \le a_{N-1} \le N \mid (a_k \in \mathbb{N})\}^2$$

and

$$\#\{0 \le a_0 \le a_1 \le \cdots \le a_{N-1} \le N \mid (a_k \in \mathbb{N})\}$$
$$= \{(p_1, \ldots, p_{N+1}) \in \mathbb{N}^{N+1} \mid p_1 + \cdots + p_{N+1} = N\}$$
$$= \binom{2N}{N},$$

the covering number for the class of functions in $\mathcal{I}$ is bounded by $\binom{2n}{n}^2$. Consider sums of powers of binomial coefficients: $a_n^r = \sum_{k=0}^n \binom{n}{k}^r$. For $r = 2$, the closed-form solution is given by

$$a_n^{(2)} = \binom{2n}{n}$$

i.e., the central binomial coefficients. $a_n^{(2)}$ obeys the recurrence relation

$$(n+1)a_{n+1}^{(2)} - (4n+2)a_n^{(2)} = 0.$$

After solving the recurrence relation we get,

$$\binom{2n}{n} = C_1 \frac{4^{n-1}}{\Gamma(n+1)} \left(\frac{3}{2}\right)_{2n-1}$$

$$((x)_n \text{ denotes Pochhammer symbol.})$$

$$= 2 \cdot \frac{2^{2(n-1)}}{\Gamma(n+1)} \left(\frac{3}{2}\right)_{2n-1}$$

(since $C_1 = 2$, which we can

obtain by setting $n = 0$ in previous equation.)

$$= \frac{2^{2(n-1)}}{\Gamma(n+1)} \frac{\Gamma(\frac{3}{2} + n - 1)}{\Gamma(\frac{3}{2})}$$

$$= \frac{2^{2(n-1)}}{\Gamma(n+1)} \frac{\Gamma(n + \frac{1}{2})}{\sqrt{\frac{\pi}{2}}}$$

$$= \frac{2^{2(n-1)}}{\sqrt{\frac{\pi}{2}}} \frac{\Gamma(n + \frac{1}{2})}{\Gamma(n+1)}$$

$$= \frac{2^{2n}}{\sqrt{\pi}} \frac{\Gamma(n + \frac{1}{2})}{\Gamma(n+1)}$$

$$\leq \frac{2^{2n}}{\sqrt{\pi}} \frac{1}{\sqrt{n}} \text{(using Lemma 3.3)}$$

$$= \frac{2^{2n}}{\sqrt{n\pi}}.$$

$$\implies \binom{2n}{n}^2 \leq \frac{2^{4n}}{n\pi}$$

$$\leq \frac{2^{4n}}{6\pi} \text{(if } n \geq 6)$$

$$\leq \frac{2^{4n}}{18}.$$

Let $n = \left\lceil \frac{LV}{\tau} \right\rceil + 1$, then

$$N(\tau, \mathcal{I}, L_2(P_n)) \leq \frac{2^{4\frac{LV}{\tau}}}{18}.$$

$\square$

### A.3 Proof of Lemma 5.7

*Proof.* Since,

$$N(\tau, \mathcal{B}', L_2(P_n)) \leq \frac{2^{16\frac{LV}{\tau}}}{324}.$$

For $z \in \mathbb{R}^d$ ,

$$N(\tau, \mathcal{B}', L_2(P_n)) \leq \left(\frac{2^{16 \frac{LV\sqrt{d}}{\tau}}}{324}\right)^d$$

Observe that,

$$\sqrt{\log N(\tau, \mathcal{B}', L_2(P_n))} \leq \frac{4\sqrt{LVd^{\frac{3}{2}}\log 2}}{\sqrt{\tau}} = g(\tau).$$

Therefore,

$$\int_a^b g(\tau)d\tau = 8\sqrt{LVd^{\frac{3}{2}}\log 2}\left[\sqrt{b} - \sqrt{a}\right] \tag{A.21}$$

We know that from Lemma (3.8) that empirical Rademacher Complexity $\hat{\mathcal{R}}_n(\mathcal{B}')$ has the following bound

$$\hat{\mathcal{R}}_n(\mathcal{B}') \leq \inf_{\epsilon \geq 0}\left\{4\epsilon + 12\int_\epsilon^b \sqrt{\frac{\log N(\tau, \mathcal{B}', L_2(P_n))}{n}}d\tau\right\},$$

where $b = \sup_{f \in \mathcal{B}'} \sqrt{\mathbb{E}[f^2]}$. Using (A.21), we have

$$\hat{\mathcal{R}}_n(\mathcal{B}') \leq \inf_{\epsilon \geq 0}\left\{4\epsilon + \frac{96\sqrt{LVd^{\frac{3}{2}}\log 2}}{\sqrt{n}}\left[\sqrt{b} - \sqrt{\epsilon}\right]\right\}.$$

This implies

$$\hat{\mathcal{R}}_n(\mathcal{B}') \leq 96\frac{\sqrt{bLVd^{\frac{3}{2}}\log 2}}{\sqrt{n}} - 576\frac{LVd^{\frac{3}{2}}\log 2}{n}.$$

$\square$

## A.4 COMPARISON WITH OTHER BOUNDS (NEURAL ODE)

**Theorem A.1.** *( Marion (2024) Generalization bound for parameterized ODEs).*

$$H_0 = x,$$
$$dH_t = \sum_{i=1}^m \theta_i(t) f_i(H_t)\, dt,$$
$$F_\theta(x) = H_1, \quad where,$$

- *$\theta = (\theta_1, \ldots, \theta_m)$ is a parameter function mapping $[0, 1]$ to $\mathbb{R}^m$.*

- *$f_i(H_t)$ represents the dynamics associated with the $i$-th component of the system.*

- *$H_t$ is the state of the system at time $t$, with the initial state $H_0 = x$.*

- *$F_\theta(x)$ denotes the output state $H_1$ after the evolution.*

$$\Theta = \{\theta : [0, 1] \to \mathbb{R}^m \mid \|\theta\|_{1,\infty} \leq R_\Theta \text{ and } \theta_i \text{ is } K_\Theta\text{-Lipschitz for } i \in \{1, \ldots, m\}\}.$$
*Consider the class of parameterized ODEs $\mathcal{F}_\Theta = \{F_\theta, \theta \in \Theta\}$. Let $\delta > 0$, then, for $n \geq 9\max(m^{-2}R_\Theta^{-2}, 1)$, with probability at least $1 - \delta$,*

$$R(\hat{\theta}_n) \leq R_n(\hat{\theta}_n) + B\sqrt{\frac{(m+1)\log(R_\Theta mn)}{n}} + \frac{Bm\sqrt{K_\Theta}}{n^{1/4}} + \frac{B\sqrt{\log\frac{1}{\delta}}}{\sqrt{n}},$$

*where $B$ is a constant depending on $K_\ell$, $K_f$, $R_W$, $R_X$, $R_Y$, and $M$. More precisely,*

$$B = 6K_\ell K_f \exp(K_f R_\Theta)\left(R_X + MR_\Theta \exp(K_f R_\Theta) + R_Y\right).$$

**Theorem A.2.** *Bleistein & Guilloux (2023) Let $G_\psi(z)$ be the dynamic function as neural network*

$$G_\psi(z) = \sigma\Big(A_q \sigma\Big(A_{q-1}\sigma\big(\cdots\sigma(A_1 z + b_1)\big) + b_{q-1}\Big) + b_q\Big), \quad \text{where} \qquad (A.22)$$

*(i) The activation function $\sigma$ is $L_\sigma$-Lipschitz. This means that for any $x, y \in \mathbb{R}$:*

$$|\sigma(x) - \sigma(y)| \le L_\sigma |x - y|.$$

*Moreover, $\sigma(0) = 0$, ensuring that the activation function is centered.*

*(ii) $A_1, A_2, \ldots, A_q$ are weight matrices for each layer of the network, and $b_1, b_2, \ldots, b_q$ are bias vectors for each layer of the network.*

*(iv) $z$ is the input vector to the multi-layer perceptron (MLP).*

*With probability at least $1 - \delta$, the generalization error $R_D(\hat{f}_D) - R_n(\hat{f}_D)$ is upper bounded by*

$$24 \frac{M_\Theta^D L_\ell}{\sqrt{2}} \sqrt{2p U_1^D + (q-1)p(p+1)U_2^D + dp(2+p)U_3^D} + M_\ell \sqrt{\frac{\log(1/\delta)}{2n}}, \quad \text{with}$$

$$U_1^D := \log(\sqrt{n} C_q K_1^D), \quad U_2^D := \log(\sqrt{np} C_q K_2^D), \quad U_3^D := \log(\sqrt{ndp} C_q K_2^D),$$

*and $C_q := (8q + 12)$. Here, $\|A_h\| \le B_A$, $\|b_h\| \le B_b$, $\|U\| \le B_U$, $\|v\| \le B_v$, $\|\Phi\| \le B_\Phi$, $K_1^D$ and $K_2^D$ are two discretization and depth-dependent constants equal to*

$$K_1^D := \max\{B_\Phi M_\Theta^D, B_v C_v\}, \quad K_2^D := \max\{B_b C_b^D, B_A C_A^D, B_U C_U\},$$

*where $C_A^D$, $C_b^D$, $C_v$, and $C_U$ are Lipschitz constants.*

$$M_\Theta := B_\Phi L_\sigma \exp(B_A L_\sigma) q L_x \Big(B_U B_x + B_v + \kappa_\Theta(0) L_x\Big),$$

$$C_A := B_\Phi L_x \exp(L_\sigma B_A) q L_x \times \max_{z \in \Omega, 1 \le i \le q} C_A^i(z), \quad C_b := B_\Phi L_x \exp(L_\sigma B_A) q L_x \times \max_{1 \le i \le q} C_b^i,$$

$$C_U := B_\Phi B_x \exp(L_\sigma B_A L_x) L_\sigma, \quad C_v := B_\Phi \exp(L_\sigma B_A).$$

$$\kappa_\Theta(0) = \frac{L_\sigma B_b}{L_\sigma B_A}\Big(q - 1\Big) L_\sigma B_A^{-1},$$

*which serves as an upper bound for $\|G_\psi(0)\|_{op}$, defined as:*

$$\|G_\psi(0)\|_{op} := \max_{\|u\|=1} \|G_\psi(u)\|.$$

## A.5 Comparisons with other bounds (Residual Neural Networks)

**Theorem A.3.** *(Marion (2024) Let $H_0 = x$, $H_{k+1} = H_k + \frac{1}{L} W_{k+1} \sigma(H_k)$, $0 \le k \le L - 1$, $F_W(x) = H_L$,*

$$\mathcal{W} = \{W \in \mathbb{R}^{L \times d \times d}, \|W\|_{1,1,\infty} \le R_W, \|W_{k+1} - W_k\|_\infty \le \frac{K_W}{L}, \quad \text{for } 1 \le k \le L - 1.\}$$

*Consider the class of neural networks $\mathcal{F}_W = \{F_W, W \in \mathcal{W}\}$. Let $\delta > 0$. Then, for $n \ge 9 R_W^{-1} \max(d^{-4} R_W^{-1}, 1)$, with probability at least $1 - \delta$,*

$$R(\hat{W}_n) \le R_n(\hat{W}_n) + B(d+1)\sqrt{\frac{\log(R_W dn)}{n}} + \frac{Bd^2 \sqrt{K_W}}{n^{1/4}} + \frac{B\sqrt{\log \frac{1}{\delta}}}{\sqrt{n}},$$

*where $B$ is a constant depending on $K_\ell$, $K_\sigma$, $R_W$, $R_X$, and $R_Y$. More precisely,*

$$B = 6\sqrt{2} K_\ell \max\left\{\frac{\exp(K_\sigma R_W)}{R_W}, 1\right\} (R_X \exp(K_\sigma R_W) + R_Y).$$

For this case also the bound by this is stricter in terms of $n$. Bound by Marion does not depend on depth but has worse dependence on width, this bound depends on depth but does not depend on width.

**Corollary A.4.** *(Marion (2024)Corollary of Theorem 1.1 of Bartlett et al. (2017b) Consider the class of neural networks $\mathcal{F}_{\tilde{W}} = \{F_W, W \in \tilde{W}\}$, where $F_W$ is given by (10) and*

$$\tilde{W} = \{W \in \mathbb{R}^{L \times d \times d} : \|W\|_{1,1,\infty} \leq R_W\}.$$

*Assume that $L \geq R_W$ and $K_\sigma = 1$, and let $\gamma, \delta > 0$. Consider $(x, y), (x_1, y_1), \ldots, (x_n, y_n)$ drawn i.i.d. from any probability distribution over $\mathbb{R}^d \times \{1, \ldots, d\}$ such that almost surely $\|x\| \leq R_X$.*

*Then, with probability at least $1 - \delta$, for every $W \in \tilde{W}$,*

$$\mathbb{P}\left(\arg\max_{1 \leq j \leq d} F_W(x)_j \neq y\right) \leq R_n(W) + C\frac{R_X R_W \exp(R_W)\log(d)\sqrt{L}}{\gamma\sqrt{n}} + \frac{C\sqrt{\log(1/\delta)}}{\sqrt{n}},$$

*where*

$$R_n(W) \leq \frac{1}{n}\sum_{i=1}^{n}\mathbb{1}_{(F_W(x_i)_{y_i} \leq \gamma + \max_{j \neq y_i} F_W(x_i)_j)},$$

*and $C$ is a universal constant.*

# B EXPERIMENT DETAILS

## B.1 FOR EXPERIMENT ILLUSTRATED BY FIGURE 1

The objective of this experiment is to analyze the effect of the number of hidden units on the generalization error of a Neural ODE model. The generalization error is defined as the model's performance on unseen test data, measured using the mean squared error (MSE). The study investigates the relationship between model complexity, as determined by the number of hidden units, and its ability to generalize.

The dataset is synthetically generated and consists of training and testing samples. The training set comprises 100 samples, while the test set includes 30 samples. Each input sample has two features, sampled from a standard normal distribution. The target values are computed using a non-linear function of the inputs with some added randomness. This introduces a non-linear relationship between inputs and targets, mimicking the challenges of real-world data.

The Neural ODE model used in this experiment consists of three main components. First, a linear input layer maps the input data into a higher-dimensional space determined by the number of hidden units. Second, the ODE function models the dynamics of the hidden state using a fully connected layer with ReLU activation, solving the ODE using the 'torchdiffeq.odeint' solver over the time interval $[0.0, 1.0]$. The final state of the ODE solver is passed through an output layer to produce the scalar prediction.

The independent variable in this study is the number of hidden units, which is varied across the following values: $[100, 200, 300, 400, 500, 600, 700, 800, 900]$. For each configuration, the model is trained for 100 epochs using the Adam optimizer with a learning rate of 0.01. The loss function used is the mean squared error (MSE), and the training process is conducted on a GPU if available. The dependent variable is the generalization error, which is evaluated as the mean squared error on the test dataset.

Reproducibility is ensured by setting random seeds for both `torch` and `numpy`. The model performance is evaluated by calculating the MSE on the training and test datasets after training. The generalization error is analyzed as a function of the number of hidden units, and a line plot is generated to visualize this relationship. The x-axis represents the number of hidden units, and the y-axis represents the corresponding generalization error.

The hypothesis of the experiment is that increasing the number of hidden units will initially reduce the generalization error as the model's capacity improves. However, beyond a certain point, overfitting may occur, leading to an increase in the generalization error. The experiment is designed to identify this trend and explore the optimal model complexity for the given task.

### B.1.1 DATA GENERATION

The dataset used in this experiment is synthetically generated to test the Neural ODE model's capability to generalize to unseen data. The process creates input-output pairs based on random input vectors and a non-linear transformation for the target values. This ensures that the task is sufficiently challenging while allowing for reproducibility.

The steps for generating the data are as follows:

1. The input data, denoted as $X$, is a set of $n_{\text{samples}}$ random vectors, where each vector has a dimensionality of input_dim. The elements of $X$ are drawn from a standard normal distribution:
$$X \sim \mathcal{N}(0, 1)^{n_{\text{samples}} \times \text{input\_dim}}.$$

2. The target values, denoted as $y$, are generated by applying a sinusoidal transformation to the sum of the elements in each input vector:

$$y_i = \sin\left(\sum_{j=1}^{\text{input\_dim}} X_{ij}\right), \quad \forall i \in \{1, 2, \ldots, n_{\text{samples}}\}.$$

This non-linear transformation introduces complexity into the data while ensuring a bounded range for the output values.

3. The inputs $X$ and the corresponding targets $y$ are paired together to form the dataset:

$$\text{Dataset} = \{(X_i, y_i)\}_{i=1}^{n_{\text{samples}}}.$$

4. Two datasets are generated:
   - A training dataset with $n_{\text{samples}} = 100$.
   - A testing dataset with $n_{\text{samples}} = 30$.

   Both datasets are created independently using the same generation process to ensure the test data remains unseen during training.

5. The generated data is stored as PyTorch tensors, making it compatible with the Neural ODE model. This enables efficient data loading and processing during training and evaluation.

This synthetic data generation process provides a controlled setup for evaluating the generalization capabilities of the Neural ODE model. The use of a sinusoidal target function introduces a non-trivial learning problem while maintaining interpretability and ease of reproducibility.

### B.2 FOR EXPERIMENT ILLUSTRATED BY FIGURE 2

The primary objective of this experiment is to evaluate the relationship between regularization parameters and the generalization gap of a Neural ODE model. The model is trained on a real-life complex dataset, simulating the trajectory of particles in a potential field. The generalization gap is computed as the difference between the mean squared error (MSE) on the training and testing datasets.

The dataset is generated synthetically to simulate realistic particle trajectories. For each trajectory, the data points are computed by introducing sinusoidal patterns with added Gaussian noise. Specifically, the $x$-coordinates are defined as $\sin(t) + 0.5\epsilon_x$, and the $y$-coordinates as $\cos(t) + 0.5\epsilon_y$, where $\epsilon_x$ and $\epsilon_y$ represent Gaussian noise. A total of 2000 samples are generated for both the training and testing datasets. The generated data is converted into tensors for compatibility with PyTorch operations.

The Neural ODE model used in the experiment consists of a multi-layer neural network. The ODE function is parameterized by a deep neural network with four fully connected layers, each containing 100 hidden units and ReLU activation. The integration of the ODE is performed using the 'torchdiffeq.odeint' solver over the time interval $[0.0, 1.0]$. Regularization is applied by augmenting the loss function with a penalty term proportional to the bound $V$, derived from the norms of the parameters and Lipschitz constants of the network layers.

The experiment is conducted for four regularization parameters: 0.0, 0.01, 0.08, and 0.1. For each parameter, 20 trials are performed, and the model is trained for 25 epochs using the Adam optimizer with a learning rate of 0.01. During each training step, the gradients of the weights are monitored to compute the spectral norm and other components required for $V$.

The generalization gap is computed as the difference between the training and test losses, both evaluated as MSE. To ensure reproducibility, the random seed is fixed at 100 for PyTorch and NumPy operations. After the training process, the generalization gaps for all regularization parameters are plotted using a box plot to visualize their distributions. The x-axis represents the regularization parameters, and the y-axis represents the generalization gaps.

This experiment highlights the trade-off between regularization and model generalization, offering insights into how different regularization strengths affect the performance of Neural ODEs on complex datasets.

### B.2.1 DATA GENERATION

The data used in this experiment is synthetically generated to simulate realistic particle trajectories in a potential field. The trajectories are designed to exhibit sinusoidal patterns with added Gaussian noise, capturing the complexity often found in real-life systems. This synthetic data allows for controlled experimentation while maintaining a level of realism.

To generate the data, the following procedure is employed:

1. Define a time vector $t$ that spans the interval $[0, 10]$ with $n_{\text{samples}}$ evenly spaced points. For this experiment, $n_{\text{samples}} = 2000$ is used for both training and testing datasets.

2. Compute the $x$-coordinates of the trajectory as:
$$x = \sin(t) + 0.5\epsilon_x,$$
where $\epsilon_x$ is a Gaussian noise term sampled from $\mathcal{N}(0, 1)$.

3. Similarly, compute the $y$-coordinates of the trajectory as:
$$y = \cos(t) + 0.5\epsilon_y,$$
where $\epsilon_y$ is another independent Gaussian noise term sampled from $\mathcal{N}(0, 1)$.

4. Combine the $x$ and $y$ coordinates to form a dataset of two-dimensional points, represented as:
$$\mathbf{data} = \begin{bmatrix} x_1 & y_1 \\ x_2 & y_2 \\ \vdots & \vdots \\ x_{n_{\text{samples}}} & y_{n_{\text{samples}}} \end{bmatrix}.$$

5. Convert the generated data into PyTorch tensors for compatibility with the Neural ODE framework. The data is moved to the computation device (CPU or GPU) to optimize performance during training and evaluation.

The generated dataset exhibits variability in particle trajectories due to the added noise, introducing challenges similar to those encountered in real-world dynamical systems. This ensures that the trained Neural ODE model can learn to approximate complex behaviors while being evaluated for its ability to generalize across different initial conditions. Separate datasets are generated for training and testing, ensuring that the model is not exposed to test samples during training.

By synthesizing the data in this manner, the experiment captures the intricacies of noisy particle trajectories, providing a robust benchmark for evaluating the generalization capabilities of the Neural ODE model.

### B.3 FOR EXPERIMENT ILLUSTRATED BY FIGURE 3

This experiment investigates the impact of Lipschitz regularization on the generalization gap in Neural Ordinary Differential Equation (ODE) models. A Neural ODE model is implemented where the parameters of the ODE depend on time. The primary goal is to examine how adding a penalty

term proportional to the Lipschitz constant of the model's weights influences the generalization gap, which is defined as the difference between validation loss and training loss.

The Neural ODE model consists of an ODE function with two fully connected layers. The first layer maps 2-dimensional input data to a hidden representation of size 50 with ReLU activation. The second layer projects this representation back to a 2-dimensional output. To incorporate time-dependency, the hidden layer's output is modulated by a sine function of time, introducing a dynamic weight adjustment. The ODE is solved using the `odeint` function from the `torchdiffeq` library over a fixed time interval of $[0, 1]$.

To measure and regulate the Lipschitz constant of the model, the singular values of the weight matrices are computed. The Lipschitz constant is defined as the maximum singular value across all weight matrices. During training, the loss function combines the mean squared error (MSE) between model predictions and ground truth labels with a penalty term proportional to the Lipschitz constant. The overall loss is expressed as:

$$\text{Loss} = \text{MSE} + \lambda \cdot L,$$

where $\lambda$ is the regularization strength, and $L$ is the Lipschitz constant of weights.

The datasets for training and validation are synthetically generated. Both datasets consist of 2-dimensional samples drawn from a standard normal distribution, $\mathcal{N}(0, 1)$. The training dataset contains 100 samples, while the validation dataset contains 20 samples. The corresponding labels are generated by scaling the input data by a factor of 2, resulting in a simple linear relationship. This ensures a clear evaluation of the model's generalization capabilities.

### B.3.1 DATA GENERATION

In this experiment, the input data and corresponding labels are synthetically generated to evaluate the generalization capability of a neural ODE model with a Lipschitz constant penalty. The input data consists of random 2-dimensional points, generated independently from a standard normal distribution. Specifically, for each input data point $x = (x_1, x_2)$, both features $x_1$ and $x_2$ are independently drawn from the standard normal distribution $\mathcal{N}(0, 1)$. This ensures that the dataset contains diverse points distributed across the 2-dimensional space. The dataset used for training consists of 100 such points, and the dataset used for validation consists of 20 points.

The corresponding labels for the input data are generated by a simple linear transformation. The label for each data point $x = (x_1, x_2)$ is computed as twice the value of the input features, i.e., $y = 2 \cdot x$. This linear transformation ensures that the label is directly related to the input data, which makes it easier for the model to learn the mapping. The training labels $y_{\text{train}}$ and validation labels $y_{\text{val}}$ are computed as $y_{\text{train}} = 2 \cdot x_{\text{train}}$ and $y_{\text{val}} = 2 \cdot x_{\text{val}}$, respectively.

The dataset is randomly split into training and validation datasets. The training dataset consists of 100 data points, and the validation dataset contains 20 data points. This splitting is done to ensure that the model is evaluated on unseen data, allowing for the measurement of its generalization performance.

To summarize, the input data is generated by independently sampling 2-dimensional points from a standard normal distribution, ensuring a variety of input values. The corresponding labels are generated through a simple linear scaling by a factor of 2. The dataset is split into training and validation sets, with 100 samples for training and 20 samples for validation. This dataset setup serves to evaluate the performance of a neural ODE model with a Lipschitz penalty term.

