# OpenReview forum: "Generalization  Bounds for  Neural Ordinary Differential Equations and Residual Neural Networks"
_ICLR.cc/2025/Conference — ICLR 2025 Conference Withdrawn Submission_

### Official Review · Reviewer_e326 · 2024-10-29

**Soundness:** 1
**Presentation:** 1
**Contribution:** 1
**Rating:** 1
**Confidence:** 4

**Summary:**

The authors claim to prove new generalization bounds for neural ODEs and residual neural networks. However, these claims are largely unsupported since their work does not significantly improve on Marion (2023) and Bleistein and Guilloux (2024). Some lemmas and proofs are directly borrowed along with notations from these two works without sufficient citations, which might be considered as a case of light plagiarism. The title is almost identical to the work of Marion (2023).

**Strengths:**

I do not believe that this paper has significant strengths.

**Weaknesses:**

**Section 2.** The related work is close to insufficient and misses several recent contributions to the field, such as Marion, Wu et al. (2024) and Chen (2024).

**Section 3.** This section compiles a list of definitions and Lemmas without sufficiently motivating their introduction. I would suggest a major rewriting of this section in order to guide the reader through the proofs.
* Also, Lemma 3.8 does not exist in Bartlett 2017b, and I believe that the Lemma as stated is wrong: there should not be a factor $1/\sqrt{n}$ in the integral, but rather a factor $1/\varepsilon$. See Bartlett 2017b Lemma A.5.

**Section 4.**
* The learning setup is unclear. The authors write "Let z be the solution of Neural ODE with x as the initial condition and let y be the true solution of the true differential equation learned by Neural ODE given by equation (4.3)" (l. 221-223): it is unclear what is meant by the "true solution". Do the authors assume a generative model for the data ? In this case, it should be introduced.
* The authors write that the empirical risk "cannot be optimized, since we do not have access to the continuous data." (l. 229 - 230). The authors have not introduced any form of continuous data, nor do they explain why the empirical risk cannot be optimized.
* This section seems to plagiarize Bleistein & Guilloux (2024) section 3.2, who consider a generative model where a continuous function, which is only observed at a discrete set of sampling times, generates the outcome through an unknown neural ODE. I believe that the authors have carelessly copied this text, hence introducing the two confusing sentences mentioned above which make no sens in their setting as it stands.

**Section 5.**

* The main contribution of this part seems to be an adaptation of Proposition 2 of Marion (2023) to the general non-linear case. The results in the section strongly resemble the results of Bleistein and Guilloux (2024) --- see Lemma 3.3. These results should be in my opinion at least ackowledged in the main text.
* Settings concerns about plagiarism aside, the result cited here is directly implied by the results from Bleistein & Guilloux (2024), who establish generalization bounds for neural CDEs of the form  $ dz(t)  = \mathbf{G}(z(t))dx(t)$ , where $ \mathbf{G} $ is a generic neural network. Indeed, by setting $ x(t) = t $, one recovers a generic neural ODE. The authors of the aforementioned paper highlight the proximity between both models in Figure 2 of their paper.
* In the abstract, the authors claim to have "showed the effect of overparameterization and domain bound in the generalization error bound". This is a strong overstatement, since the type of arguments used by the authors only work in the case where $n$ is taken to be sufficiently large to obtain concentration ; even if in this case the number of parameters exceeds the number of observations, these bounds become vacuous in this setting, since the bound presented in Theorem 5.9 does not tend to $0$ when $d$ grows at the same rate than $n$. Hence these bounds say nothing about the overparametrized regime, in which it is typically observed that neural network achieve good prediction performance even if they completely overfit the training data (see for instance Bartlett 2019).

**Section 6.**
* Both Marion (2023) and Bleistein and Guilloux (2024) invoke discretization based arguments to go from continuous neural-ODE like architectures to discrete ResNet-type architectures. I do not see such an argument here, and am hence unconvinced by the soundness of Theorem 6.1. In particular, the authors simply write that "a neural ODE with an euler solver and $\Delta t= 1$ replicated the ResNet updates, it follows that the solution space of ResNets is contained within the solution space of Neural ODEs." (l. 366-368).

**Section 7.**

* The authors claim to perform these experiments on neural ODEs. Given the previous approximations in the paper and the strong similarities of experiments displayed in figures 2 and 3 with the experimental section of Marion (2023), I strongly suspect that these experiments are carried out on ResNets rather than **continuous** neural ODEs.
* Writing that experiment 1 validates Theorem 5.9 (l. 385) is an overstatement: the authors show (without any confidence intervals) on a purely synthetic dataset that the generalization error increases with the number of hidden units. However, the details on the model are insufficient (what is exactly meant by the number of hidden units ?). Also, since no precision is given on the training data, it is unclear whether the model operates in an overparametrized or underparametrized setting.
* The experiment displayed in Figure 3 is directly copied from Marion (2023). This article should at least be acknowledged here in my opinion.
* The experiment displayed in Figure 2 is novel and investigates the effect of penalizing the loss of a neural ODE with the bound of the solution, hence favoring solutions with a low euclidian norm. However, the experiment is not conclusive due to the high variance and the little variability of the mean for every choice of regularization.
* Figures should be included in a vectorized format (PNG or PDF).
* A experimental appendix should be added, that includes a detailed overview of the experiments.

**References**

Bartlett, Peter et al., "Spectrally-normalized margin bounds for neural networks", Neurips 2017.

Bartlett, Peter et al., "Benign overfitting in Linear Regression", Proceedings of the National Academy of Sciences, 2020.

Bleistein, Linus and Guilloux, Agathe, "On the Generalization and Approximation Capacities of Neural Controlled Differential Equations", ICLR 2024.

Chen, Yihang et al., Generalization of Scaled Deep ResNets in the Mean-Field Regime, ICLR 2024.

Marion, Pierre, "Generalization bounds for neural ordinary differential equations and deep residual networks", Neurips 2023.

Marion, Pierre, Wu, Yu-Han, et al., Implicit regularization of deep residual networks towards neural ODEs, ICLR 2024.

**Questions:**

I believe that this paper is largely insufficient for publication as it stands due to a lack of novelty, and often teeters on the brink of plagiarism. I strongly encourage the authors not to submit this work at the moment and to read the ethics requirements of ICLR 2025. Many points can be improved (see above). I list a few questions bellow.

* Can you provide extensive details on the experiments carried out ? In particular, I would appreciate a mathematical formulation of the model use to generate your data and architectural details. Also, please carefully check that the experiments are run with neural ODEs instead of ResNets.
* Please provide more mathematical details on your neural ODE to ResNet conversion (Theorem 6.1).

**Details Of Ethics Concerns:**

This paper seems to partially plagiarize Marion (2023) and Bleistein and Guilloux (2024), without any significant contribution. I have provided more details above.

---

> ### Author Response · Authors · 2024-11-18
>
> Answers for Weakness.
>
> Section 2.
> We have already referenced them in Abstract and Introduction.
>
> Section 3.
> We have used a variation of it  link: https://www.cs.cornell.edu/~sridharan/dudley.pdf
>
> Section 4.
> The learning setup is similar to that of  [1]. Sorry for confusion, we meant true real value(not function) not true solution at final time, we used generative model with unknown ODE.
> How the data is generated is explained in answer given below to question 1.
>
>  Since we used generative model with unknown ODE, we do not have access to continuous data.
>
> Section 5.
>
> This is not the main contribution ,we will still have generalization  bound if we do not  use this result , We found stricter bound for the covering number of solution space of Neural ODE, this is main contribution .We used this to get the bound of solution in terms of norm of weights ,We have not used directly any Lemma or Proof from Marion [2] or Blesistein and Guilloux [1]
>
> Yes, by setting $x(t) =t$, one recovers a generic neural ODE, but their  bound depend on discretization of control variable, in this case time, also the bound consists of more number of parameters.
>
> I think there is confusion in understand from your side $d$ will not say anything about over-parameterization, $d$ is the dimension of solution, the number of parameters will have effect on Norm and $V$ is dependent on norm. In Marion's paper also it is dependent on bound on norm of weight in the numerator.
>
> Section 6.
>
> This is addressed in answer given to Question 2 below.
>
> Section 7.
>
> 1.Code is for neural ODE, so one can verify the code and rerun it to reproduce the results.
>
>  2.The number of neurons in a layers is referred as number of hidden units.details of synthetic data generation is provided,it is actually real life complex data for particle moving in the air.
>
>  3. We have acknowledged the work of Marion [2].
>
>  4.Yes ,but the mean is either same or it is decreasing although variance is high.If one zooms the image one can find.
>
>  5. We have added figure in png format now.
>
>  6. We have added more details  in the  Appendix section.
>
> Answers to Questions.
>
> 1. The synthetic data is generated to mimic complex real-life particle motion rather than being derived from numerical solutions of an explicit ODE. Specifically, it simulates the motion of a particle in a potential field with a sinusoidal pattern plus random noise, as defined by:
> $$
> x = \sin(t) + 0.5 \cdot \text{noise}
> $$
> $$
> y = \cos(t) + 0.5 \cdot \text{noise}
> $$
>
> This noisy sinusoidal motion is intended to represent realistic, complex trajectories, suitable for testing the neural ODE model’s ability to generalize. Also, the experiments are run with neural ODE, you can verify using codes provided  in supplementary.
>
> 2. We have provided more mathematical details. We meant that the bound on time dependent Neural ODE will be same, that is when the parameters are also time dependent. The bound for time independent Neural ODE will be less than ResNet since Lipschitz constant  of weights will be zero in that case. If L(final time of neural ODE) is discrete then the final time L of neural ODE will be same as Number of layers of Resnet, $V$ will be same since it independent of time, $L$ can be real number and since the number of layers are discrete, we take $N=floor(L)$. Since $f$ is same for both $Lip(f)$ will also be same hence V will be same. $N$ will always be less than or equal to $L$, hence covering number for set of solutions by Resnet will be less than or equal to covering number for set of solutions by Neural ODE and thus the Rademacher complexity. This is why bound is same.
>
> [1] Bleistein et al. On the Generalization and Approximation Capacities of Neural Controlled Differential Equations, ICLR 2024.
>
> [2] Marion, Generalization bounds for neural ordinary differential equations and deep residual networks, NeurIPS 2023.

---

> ### Comment · Reviewer_e326 · 2024-11-22
>
> These answers do only partially alleviate my concerns with this paper. I will maintain my score.

---

### Official Review · Reviewer_BqoU · 2024-11-02

**Soundness:** 2
**Presentation:** 1
**Contribution:** 2
**Rating:** 3
**Confidence:** 3

**Summary:**

Generalization bound of neural ODEs whose vector field is an MLP

**Strengths:**

1. give an bound estimation of $z(t)$ based on the Lipschitz constant of an MLP $f(z)$
2. follow the similar technique as [1] to derive the generalization bound of $\dot z= f(z) $

[1] Bartlett et, al. Spectrally-normalized margin bounds for neural networks, NeurIPS 2017.

**Weaknesses:**

1. The bound estimation of $\|z(t)\|$ is very loose due to the Gronwall's inequality $u(t)\leq \alpha(t)+\int_a^t\beta(s)u(s)ds$. In this case, $\beta$ is the Lipschitz bound $\mathrm{Lip}(f)$ and $\alpha$ is a bound related to $\mathrm{Lip}(f)$ and bias norm. Thus, the downstream analysis of generalization bound could be very conservative.
2. $\mathrm{Lip}(f)$ is estimated using the product of spectral norm bounds, which is again very loose. The SOTA estimation is based on some semidefinite programming formulation, see [2].
3. The assumption of globally Lipschitz $f$ is quite strong as the popular transformer architecture is only locally Lipschitz.
4. The experimental results are quite weak, lacking of extensive comparison study.
5. The presentation is poor. A few (not all) examples are listed as follows:

- The mapping $z(t)\rightarrow y$ is not defined.
- Assumption 2 involves $A_i(t)$ and $b_i(t)$ which are introduced later in Section 5.
- A right ) is missing in 5.4.
- [1] appears twice in the reference.

[2] P. Pauli et. al. Novel quadratic constraints for extending lipsdp beyond slope-restricted activations, ICLR 2024.

**Questions:**

1. There exist many generalization bound analysis of residual networks. Can you provide some comparison studies with Thm. 6.1?

---

> ### Author Response · Authors · 2024-11-18
>
> Answers for Weakness.
>
> 1. Thanks for pointing this out, but our aim in this paper was to get stricter bounds for covering numbers. We derive tighter bound on covering number but for the solution  bound, we used the Lipschitz based argument.
> 2. Thanks for this interesting observation. This observation will help from the point of view of implementation, but the theoretical bound will have no effect since our bound does not have explicit form for estimation of  Lip$(f)$. Our bound assumes Lip$(f)$ is known.
> 3. Yes, you are correct, it is not  globally Lipchitz for some activation functions but we can choose activation functions to make $f$ globally Lipschitz.
>  4. The experiment's aim was not to compare in terms of  strictness of bound  because for that  we believe the bound should have same type of parameters. The experiment's aim was to get the relation of the bound of the solution and the Lipschitz constant  of weights with the generalization bound.
> 5. We have modified the manuscript based on the comments of reviewer, as in earlier version presentation was not up to the mark.
>
> Answers for Questions.
>
> 1. The bound given in our work is stricter in terms of $n$. Bound given by Marion [1] does not depend on depth but has worse dependence on width, this bound depends on depth but does not depend on width.  Same is true for Bartlett et al. [2] work. We have added comparison.
>
>
> [1] Marion Generalization bounds for neural ordinary differential equations and deep residual networks, NeurIPS 2023.
>
> [2] Bartlett et al. Spectrally-normalized margin bounds for neural networks, NeurIPS 2017.

---

> > ### Comment · Reviewer_BqoU · 2024-11-22
> >
> > Thanks for your response. I am not totally convinced yet.
> >
> > 2. For deep neural networks, it is NP-hard to compute $\mathrm{Lip}(f)$. Thus, your assumption is too strong.
> > 3. I don't think it is problem of activation functions. It is the model architecture, e.g., attention mechanism which involves quadratic terms and thus it is only locally Lipschitz.
> >
> > Thus, I tend to maintain my score.

---

> ### Author Response · Authors · 2024-11-22
>
> Thank you for the  reply.
>
>   We used the idea similar to [1] just to  get relation between bound of solutions and parameters .Even if we don't assume  f is Lipschitz(global or local) and don't use Lif(f) in the bound of solution  we will still have genralization bound consisting of V term, since the solution will be of bounded variation (it requires only condition that solution is differentiable). One can find V in many different ways.
>
> 2.Yes we agree for deep neural networks, it is NP-hard to compute Lip(f), but prior work[1] also has Lip(f) in their bound (equation 120  Lip(G) term ). It does not describe its estimation
>
> 3.Yes, it depends not only on activation functions but also on other factors such as weights and overall architecture. However, the prior work [1] done in this area for general function does not assume it is locally Lipschitz ( section 3.1  Neural Vector Fields first line).
>
>
> [1] Bleistein et al. On the Generalization and Approximation Capacities of Neural Controlled Differential Equations, ICLR 2024.

---

### Official Review · Reviewer_SUqg · 2024-11-03

**Soundness:** 2
**Presentation:** 2
**Contribution:** 3
**Rating:** 5
**Confidence:** 3

**Summary:**

This paper provides generalization bounds for neural ODEs. It extends the class of neural networks used as the dynamics function. The bound is applied also to residual neural networks. Numerical experiments are used to show the effect of hyperparameters on the generalization gap.

**Strengths:**

* The results are applicable to a much larger class of neural ODEs than prior work, such as Marion's paper which is only for when f depends linearly on the parameters.
* The prior work is well explained. Important lemmas from the prior work as used.
* I did not find errors in any of the proofs themselves.

**Weaknesses:**

* Some of the notation is conflicting/confusing. Please see the questions.
* The prior work of Bleistein and Guilloux, and Marion is referenced. However, the bounds derived in these paper and in this paper are not compared.
* The main theorems 5.9 and 6.1 have only an outline of the proof, and there is not a full proof in the appendix.
* The numerically illustrations are missing details. Please see the questions.

**Questions:**

* On lines 89 and 93, the initial condition is given by $z(0)=\phi_{\theta(t)}(u)$. Why does the initial condition depend on the parameters at time t instead of time 0?
* On line 168, should it be "if" instead of "then"? Is line 169 an assumption of the lemma?
* In Assumption 3 (line 236), the outcome y is said to be in R. Is y a single number (say the solution at a final time) or is y a function of time? Same question for Assumption 4 and the loss function. Does this theory only apply for one-dimensional ODEs?
* Why is the risk a function of f in Definition 3.9, but is a function of z in section 4?
* Why is f a function of z, t, and theta in Assumption 1, but only a function of z and theta in equation (4.3)?
* In the numerical illustrations, what are the  regularization loss functions for either case?
* In the numerical illustration, how is the synethic data generated? Is it from numerical solutions of an ODE? If so, what ODE?

---

> ### Author Response · Authors · 2024-11-18
>
> Answers for Weakness.
>
> 1.  We have changed some of the notations for the convenience of readers.
> 2. Our bound is stricter in terms of $n$ than Marion.  For bound given by Bleistein and Guilloux, if we take the case $x(t)=t$, in which case it is neural ode, the bound is the same in terms of $n$ but the bound depends on the discretization of time, here it is independent of that and also the bound is simpler in this case as it contains less number of parameters. Comparing the bound given by Marion, the bound in our work is stricter in terms of $n$ for the linear case. For residual neural networks, the bound given by Marion does not depend on depth and have worse dependence on width. Our bound depends on depth but does not depend on width. We have added comparison results.
> 3. This is just substitution of bound on Radamacher complexity to theorem on regression bounds by  Mohri 2018.
>
> Answers for Questions.
>
> 1. Thanks for pointing out, yes there is typo, it  should depend at time 0. We have changed it.
> 2. Yes you are correct, it should be "if" instead of "then" and line 169 is an assumption of lemma.
> 3. No, it is a vector in $\mathbb{R}^{d}$ (the solution at a final time) and not function of time. The theory will be valid for any finite-dimensional ODE (the solution can be of any dimension).
> 4. There is typo in definition, it should be function of $z$. For more details, we refer to [1], they have used the transform version of $z$.
> 5. Thanks for pointing out this, it should be function of $z, t$, and $\theta$ everywhere. We refer to [2]
> 6. We used bound on solution as regularization loss function  for figure 2 and lipschitz constant of weights  as regularization loss function for figure 3.
> 7. The ODE is unknown. The synthetic data is generated to mimic complex real-life particle motion rather than being derived from numerical solutions of an explicit ODE. Specifically, it simulates the motion of a particle in a potential field with a sinusoidal pattern plus random noise, as defined by:
> $$
> x = \sin(t) + 0.5 \cdot \text{noise}
> $$
> $$
> y = \cos(t) + 0.5 \cdot \text{noise}
> $$
>
> This noisy sinusoidal motion is intended to represent realistic, complex trajectories, suitable for testing the neural ODE model’s ability to generalize.
>
> [1] Bleistein et. al  On the Generalization and Approximation Capacities of Neural Controlled Differential Equations, ICLR 2024.
>
> [2]  Chen et. al Neural Ordinary Differential Equations, NeurIPS 2018.

---

> > ### Comment · Reviewer_SUqg · 2024-11-22
> >
> > Thank you for your response and some clarifications. However, after reading all other reviews and responses, I have decided to maintain my score.

---

> ### Author Response · Authors · 2024-11-23
>
> Please consider the new response as well given to BqoU.

---

### Official Review · Reviewer_2svr · 2024-11-05

**Soundness:** 1
**Presentation:** 1
**Contribution:** 2
**Rating:** 3
**Confidence:** 3

**Summary:**

This submission proposes generalization bounds for neural ODEs and residual neural networks (the latter being a discretization of the first).
The first 5 pages of the paper are dedicated to related works and preliminary results. The main results are in Theorems 5.9 and 6.9, generalizing results from Marion (2024), where the author only consider linear parametrization of the residuals (while still having non linear residuals with respect to the activations). Experiments on synthetic data are conducted.

**Strengths:**

The paper studies the interesting problem of deriving generalization bounds for residual architectures which are at the core of most successful deep learning methods.

**Weaknesses:**

In my opinion, this paper is not suitable for acceptance at ICLR. It appears incomplete and lacking in polish. Below are specific points:

- The paper shares almost the exact same title as Marion (2024), with only the term "deep" removed. This is inappropriate.
- The obtained generalization bounds in Theorems 5.9 and 6.1) are not commented on and, most importantly, are not compared with existing ones from Marion (2024).
- The second experiment (Fig. 2) appears very similar to experiments shown in Marion (2024) (Figs. 1 and 2), yet Marion’s work is not cited here.
- The paper references only around 20 prior works, which is insufficient. A broader acknowledgment of previous studies is necessary (see the references cited by Marion (2024) as a comparison).
- The bibliography is poorly presented and lacks formatting consistency.
- There are no experimental details provided. I looked in the appendix, but none were included.


Overall, the paper appears to have been submitted without adequate proofreading (see, for instance, the last sentence of the abstract). In addition:

- In Assumption 1, there is an unexpected dependence on time—this should be clarified.
- The symbol $z$ in line 227 is the same as the notation for the ODE solution. This is confusing.
- Line 230: The expression involving the $\arg\min$ is difficult to understand. The $\arg\min$ is taken over $\theta$, yet it is denoted as a function $f$ (that is itself parametrized by $\theta$). Furthermore, $\arg\min$ is applied to $\theta(t) \in \theta(t)$, which is extremely unclear and problematic.
- Line 235: Typo present.
- In Lemma 5.1, it would be helpful to explicitly state that Assumption 1 is being used.
- In Lemma 5.1, the structure is confusing. It is hard to tell what is an assumption and what is a result. Key definitions (e.g., for $f$) are also missing. Can you clarify ?
- Line 265: The presentation here lacks rigor. Can you please specify the assumptions?
- There are multiple typos in the use of parentheses in lines 262, 267, and 272, which is unacceptable.
- Multiple typos are present in the experimental section (e.g., lines 423 and 429).

**Questions:**

- How different are your experimental results from those of Marion (2024)?
- Why is there sometimes an additional t argument within the parametrized function in the neural ODES?
- Both theorems provide the same bounds, which requires clarification—how is this possible?
- Please also see the questions and remarks in the previous section.

---

> ### Author Response · Authors · 2024-11-18
>
> Answers for Weakness.
>
> 1. Sorry for this confusion. We have changed the title and also the method used by Marion are totally different, because he used the covering number bound for $\theta$ and our focus is on the class of solution of neural ODE itself.
> 2. The bound by this is stricter in terms of $n$. Bound by Marion does not depend on depth but has worse dependence on width, this bound depends on depth but does not depend on width.
> 3. The experiments are not similar. The regularization term is different in our case.
> 4. Thanks for the suggestion, we have added some more references.
> 5. We have updated reference formatting.
> 6. We have already added experiment details in numerical illustration itself , but still  we provide some more experimental details  such as how the data is generated  for the convenience of readers.
>
> Answers for Questions.
>
> 1. There is typo it  should be $t$ everywhere inside $f$  , because the dynamic function  is $f(z(t),t,\theta)$  for example the dynamic function can be  $z(t)t^2\theta$, authors in [1] also used this type of notation.
> 2. Yes, it is a solution of Neural ODE. But $z$ itself will depend on $\theta$, so $\theta$ dependence is there. Also, if we see [2], the risk is defined for tranformed version of solution only i.e., $\Phi^{T}z_{1}$.
> 3. It will be $\hat{\theta}$ and $\hat{f}$ with be predictor for this $\hat{\theta}$. It should be $z_{\theta(t)}{(t)}$ which is solution to Neural ODE.
> 4. Marion is trying to show the link between the generalization gap and the Lipschitz constant of the weights, here we are also trying to show the link between the generalization gap and the bound on solution. Marion has done the experiment shown in fig 3 for Resnet, we have done for Neural ODE.
> 5. Already addressed in point 1.
> 6. We meant that the bound on time dependent Neural ODE will be same, that is when the parameters are also time dependent. The bound for time independent Neural ODE will be less than ResNet since Lipschitz constant  of weights will be zero in that case. If L(final time of neural ODE) is discrete then the final time L of neural ODE will be same as Number of layers of Resnet, $V$ will hold for any  time, it will be same for final time t for Neural ODE and Resnet, $L$ can be real number and since the number of layers is discrete we take $N=floor(L)$. Since $f$ is same for both,  $Lip(f)$ will also be same hence $V$ will be same. $N$ will always be less than  or equal to $L$, hence covering number for set of solutions by Resnet will be less than or equal to covering number for set of solutions by Neural ODE and thus the Rademacher complexity. This is why bound is same.
>
> [1]  Chen et. al Neural Ordinary Differential Equations, NeurIPS  2018.
>
> [2] Bleistein et. al  On the Generalization and Approximation Capacities of Neural Controlled Differential Equations, ICLR 2024.

---

> > ### Comment · Reviewer_2svr · 2024-11-22
> >
> > Thank you for your response to my review. I have read your answer as well as the other reviews and your answers to them.
> >
> > I decide to maintain my score to 3 as I believe this paper should not be published at the moment, for the reasons listed in my review.

---

> > > ### Author Response · Authors · 2024-11-23
> > >
> > > Thank you for the reply
> > > Can you let us know the problem with the answers?

---

> > > > ### Comment · Reviewer_2svr · 2024-11-24
> > > >
> > > > Your answers do not convince me not the reject the paper hence my decision to maintain my score. In addition, the other reviews confirm to me that the paper lacks originality in its contribution and its ressemblance to Marion (2023) worries me on the transparency of your work.

---

> ### Author Response · Authors · 2024-11-25
>
> It seems the reviewers has not thoroughly gone through the paper.None of the work either by Gulliox et. al (2024) or Marion(2023) follows the method of finding  stricter bound of covering number of bounded variation functions.Only fig 3 resembles to that of Marion(2023)  still that is for ResNet and for different dynamics to be learned and a small part(finding the relation between  bound of solution and parameters) is similar(None of the complete Lemma and Theorem were used)  to  Gulliox et.al (2024 ), but we have found for time dependent case.
>
> We are disappointed by the reviewers since none has pointed  out the main contribution.

---

### Author Response · Authors · 2024-11-27

We have uploaded the revised version of paper.
1. The title has been changed.
2. Typos, bibliography formatting corrected.
3 . We have made changes in section 4(such as x is changed with t,y is label instead of solution etc ).
4. We have made changes in section 5 (Theorem 5.9 and 6.1)
5. We have added comparisons.
6. Experimental details are added in the Appendix.

---

### Author Response · Authors · 2024-12-02

Today is the last date to reply for the reviewers but we did not get a reply.

---

### Note · Authors · 2024-12-12

I have read and agree with the venue's withdrawal policy on behalf of myself and my co-authors.